# Role of Postbiotics in Diabetes Mellitus: Current Knowledge and Future Perspectives

**DOI:** 10.3390/foods10071590

**Published:** 2021-07-08

**Authors:** Miriam Cabello-Olmo, Miriam Araña, Raquel Urtasun, Ignacio J. Encio, Miguel Barajas

**Affiliations:** Biochemistry Area, Department of Health Science, Public University of Navarre, 31008 Pamplona, Spain; miriamcabelloolmo@gmail.com (M.C.-O.); miriam.arana@unavarra.es (M.A.); raquel.urtasun@unavarra.es (R.U.); ignacio.encio@unavarra.es (I.J.E.)

**Keywords:** postbiotics, diabetes mellitus, bacteria-derived factors, bioactive compounds, functional foods, health benefits, lactic acid bacteria, gut microbiota, probiotics, paraprobiotics

## Abstract

In the last decade, the gastrointestinal microbiota has been recognised as being essential for health. Indeed, several publications have documented the suitability of probiotics, prebiotics, and symbiotics in the management of different diseases such as diabetes mellitus (DM). Advances in laboratory techniques have allowed the identification and characterisation of new biologically active molecules, referred to as “postbiotics”. Postbiotics are defined as functional bioactive compounds obtained from food-grade microorganisms that confer health benefits when administered in adequate amounts. They include cell structures, secreted molecules or metabolic by-products, and inanimate microorganisms. This heterogeneous group of molecules presents a broad range of mechanisms and may exhibit some advantages over traditional “biotics” such as probiotics and prebiotics. Owing to the growing incidence of DM worldwide and the implications of the microbiota in the disease progression, postbiotics appear to be good candidates as novel therapeutic targets. In the present review, we summarise the current knowledge about postbiotic compounds and their potential application in diabetes management. Additionally, we envision future perspectives on this topic. In summary, the results indicate that postbiotics hold promise as a potential novel therapeutic strategy for DM.

## 1. Introduction

Since ancient times, our forefathers were aware of the importance of fermented foods and fermenting microorganisms for well-being, and in particular for intestinal health [1]. At present, the evidence for the role of the gastrointestinal microbiota (GM) in the host’s health is robust, and this is because of the endocrine, digestive, and defensive functions achieved by our microbes [2,3,4]. Perturbations of the GM (dysbiosis) have been identified in several pathological states, including different forms of diabetes mellitus (DM), inflammatory bowel diseases, cancer, neurological diseases, psychiatric disorders, and even allergies, conditions in which the GM seemed to contribute, to some extent, to the disease onset and progression [5,6,7,8]. Moreover, a large number of investigations have demonstrated that strategies aimed at modulating the GM composition or activity, for instance, employing probiotics or prebiotics supplementations, are particularly useful for the restoration of the intestinal microbial environment and therefore for the host’s health condition [9,10,11]. Probiotics and prebiotics are undoubtedly the most studied GM modulators. Probiotic refers to alive microorganisms that provide health benefits to the host through several mechanisms, including improvements of the intestinal barrier function, protection against pathogens, and the modulation of the immune response [12,13]. The most commonly used are species belonging to the genera *Lactobacillus, Bifidobacterium, Streptococcus,* and *Lactococcus* among bacteria and *Saccharomyces* among yeasts [14,15,16]. Prebiotics are non-living digestive microbial stimulants that are selectively fermented by our resident microbiota [17]. Hitherto, the term was used in essence for non-digestible fibres; however, the current definition also extends to bioactive compounds of different origin, such as polyunsaturated fatty acids and polyphenols [18].

In addition to the unquestionable beneficial effects of probiotics and prebiotics, emerging concepts are opening the door to new microbiome-based approaches. Figure 1 highlights the major historical milestones in microbiology, the biotics family, and health, along with other significant landmarks. Thanks to technological process in the last decades, tremendous advancements have taken place regarding the study, classification, and characterisation of the different probiotic-related concepts. Additionally, relevant is the foundation of The International Scientific Association for Probiotics and Prebiotics (ISAPP), which helps in the definition of the emerging concepts by providing expert consensus, among other activities. In addition to the new biotics terms, the notion of postbiotics has to be emphasised, which is a term first coined in 2012 by Tsilingiri and colleges [19] and recently updated in the ISAPP expert consensus document [20]. Experimental studies have indicated that microbial components can exhibit different bioactivities than their viable counterparts (probiotic) [21], which is why they present an attractive area for research. The category of postbiotics has gained significant interest; in fact, more than 87% of total scientific publications on this topic have been carried out in the past three years (according to Pubmed.gov, cosulted in June 2021).

### 1.1. Postbiotics

Postbiotics refer to probiotic-derived products obtained from food-grade microorganisms that confer health benefits when administered in adequate amounts [22]. They include cell structures and secreted products or metabolic by-products that are discharged by viable microbial cells or that are collected and isolated after the cell lysis [23]. Most postbiotics are derived from bacteria, commonly *Lactobacilli* and *Bifidobacterium* members; however, fungal origin postbiotics are also investigated [24]. The newly published ISAPP postbiotic definition [20], “preparation of inanimate microorganisms and/or their components that confers a health benefit on the host”, also includes inanimate microorganisms. At present, there are some commercial postbiotics, in the form of supplements or incorporated in food matrices, that are mostly intended for their use in gastrointestinal or immune-related pathologies [21]. Findings from previous studies are encouraging, and the future of postbiotics in other clinical applications, including different forms of DM [25,26], seems promising and is the subject of this review.

#### 1.1.1. Characteristics of Postbiotics

The growing interest regarding postbiotics is partially explained by the advantages they provide, which are summarised in Figure 2. Briefly, (i) they are quite safe; (ii) they are well-tolerated and associated with reduced risk for adverse effects in vulnerable individuals (pregnant women, premature children, old adults, subjects immunocompromised or presenting an impaired immune function or intestinal barrier) [27,28,29,30]; (iii) they have no risk for transferring antibiotic-resistant genes to pathogenic or commensal bacteria [31]; (iv) their effectivity is independent of the cell viability, which ensures longer stability and improved shelf-life [20]; (v) they present an easy industrial (large-scale) production [32]; (vi) they show interesting technological properties (i.e., rheological properties of exopolysaccharides (EPS) in the food industry as a stabiliser [32,33] or bio-preservative effects of LAB bacteriocins [34]). Other properties include (vii) the wide range of health-promoting effects they provide (See Figure 3), some of which can be reinforced in comparison with the effect of intact viable microbial cells [35]. Another very interesting feature of postbiotics is that, due to their nature, (viii) it appears feasible that they could be used with concurrent administration with antibiotic and antifungal agents.

#### 1.1.2. Classification

The group of postbiotics covers a plethora of compounds. First, we find microbial-derived metabolites, such as enzymes, proteins, peptides, organic acids, vitamins, minerals, bacteriocins, antimicrobial peptides, and extracellular vesicles (EVs) [31,34], along with other excreted molecules such as EPS [36]. Second, we find other molecules that shape the cell structure, including cell wall components, such as the polymers teichoic acids (lipoteichoic and wall teichoic acids), peptidoglycans, peptidoglycan-derived muropeptides, pili-type forms, and cell surface fractions (i.e., S-layer protein, mucus binding protein, fibronectin-binding protein) [23,32]. Third, other postbiotics are defined as cell-free extracts and lysates, culture supernatants, or biosurfactants (cell-wall-associated or intracellular) [24,31]. Last, as previously mentioned, inanimate microorganisms may also be considered in the postbiotic category [20,37].

The techniques utilised for the extraction and purification of postbiotics are many and varied. They must engage the microbial membrane and disturb the cell integrity to collect the intracellular content [21]. To mention some examples, solvent extraction, deproteinisation and precipitation, separation by electrophoresis and analysis with liquid chromatography or sonication, and hydrophobic gradient chromatography are among the applied techniques (reviewed in detail in [32]). Regarding microbial inactivation, different procedures or techniques can be used, including heat, high pressure, irradiation, or sonication [37,38].

#### 1.1.3. Health Benefits of Postbiotics

Since the concept of postbiotic involves a broad range of compounds and substances diverse in nature and content (mentioned above), very different biological effects have been described (Figure 3). Some of these effects are restricted to the intestinal tract, for example, establishing a healthier GM [21,23], exerting prebiotics effect [36], or controlling the gut permeability [39,40]. Despite this, other effects not only affect the epithelial barrier but also cause systemic effects [24]. Collectively, the current evidence indicates that postbiotics can exert anti-inflammatory, antioxidant, immunomodulatory, pathogen inhibitory, anti-obesogenic, anticancer, antitumor, antiproliferative, antibiofilm, anti-adhesion, antihypertensive, hypocholesterolaemic, hepatoprotective, cardioprotective, anti-atherosclerotic, and anti-ulcerative effects (previously reviewed in [23,24,31,32]).

It is important to emphasise that, similar to probiotics [18], it is likely that postbiotics properties are species- and strain-specific and depend on the microbial progenitor utilised for their formulation [21,41]. In addition to this, the type and activity of the resulting postbiotic will also depend on the substrate or matrix where the postbiotic compounds are produced [42].

The GM is probably the major source of postbiotic constituents. Our resident microbes release a myriad of products, including metabolites and also cell components (reviewed in detail in [43]), that behave as messengers in the microbiota–host interactions and are of major significance to the host [44,45]. In the present review, however, we focus on the potential of exogenous factors that are orally administered.

Finally, in addition to their use in the food industry and clinical applications, postbiotics have also been exploited in activities as varied as their use in animal health [46,47] or sport performance [48].

### 1.2. Lactic Acid Bacteria

Lactic acid bacteria (LAB), such as *Lactobacillus, Bifidobacterium,* and *Pediococcus spp*., are without doubt the most important bacteria group in the food industry. They can be homofermentative or heterofermentative, depending upon their carbohydrate metabolism [36]. LABs have a long history of safe use [49] and are widely used in food processing, where they play important roles [50,51]. They act as cell factories and produce a plethora of potentially bioactive compounds including functional EPSs [33,52], enzymes [53], vitamins [54], anti-inflammatory substances [55], peptides [42], and antimicrobial products [34]. All of them show interesting technological attributes for food production and preservation, as well as exerting beneficial effects on human health, as described below.

### 1.3. Microbiotherapy for Diabetes Mellitus Management

Diabetes mellitus is a chronic disease characterised by elevated blood glucose levels (hyperglycaemia), originated by an autoimmune β-cell destruction (type 1 diabetes mellitus, T1DM) or a progressive loss of pancreatic function due to inadequate insulin secretion by the β cells as a consequence of insulin resistance exerted by peripheral tissues such as liver, muscle, and adipose tissue (type 2 diabetes mellitus, T2DM) [56]. The epidemiological evidence indicates that DM’s prevalence is reaching worrisome levels, and this scenario is exacerbated by poor diet quality, sedentarism, obesity pandemic, and the growing population aging, among other factors. The picture is alarming since DM is a major cause of death worldwide, and diabetes-specific complications are a leading cause of disability, particularly cardiovascular complications [57,58].

During recent years, evidence has accumulated to support that the resident GM is among the set of environmental factors implicated in DM development and progression [59,60]. Possible explanations include the symbiotic relationship between the host and their intestinal microorganisms. The role of the resident microbes in the host’s energy balance, metabolism, inflammation, and immunity is widely accepted [4,61,62,63]. The former are particularly interesting in the case of T2D, the latter is more involved in T1D development, and inflammation is a common factor between both forms of DM [64]. Additionally, the GM can also influence the host’s homeostasis through other functions such as nutrient absorption, intestinal permeability, or controlling gene expression [65]. In connection with DM, findings from experimental studies suggest that specific bacterial components could foster (LPS) [66] or protect (peptidoglycans) [67] from its onset or development. Furthermore, certain bacteria groups have been associated with improved glucose metabolism [66] or have been correlated with fasting blood glucose, HbA1c, or even insulin levels [67]. Moreover, some GM derived-metabolites (i.e., bile acids, indole, short-chain fatty acids (SCFAs)) could either directly or indirectly modulate energy homeostasis and glucose metabolism [67].

Findings from cross-sectional and animal studies have shown that diabetic GM presents an unfriendly composition and activity, and some features are shared among T1D and T2D patients [68]. This “diabetic microbiota” includes a loss of butyrate-producing species, enrichment of opportunistic microorganisms, an overall lower gene count (abundance) associated with metabolic impairments, changes in nutrients’ transport, enzymatic activity, and metabolism, all of this affecting SCFA concentrations and oxidative stress response, among others [66,67,69]. Moreover, diabetic patients commonly present gastrointestinal alternations (i.e., alteration of the bowel movement frequency) and have an increased risk for giving some gastrointestinal disorders [70]. DM has been linked to altered intestinal permeability, and a leaky gut has been reported in both T1D [71] and T2D [72]. This context favours the occurrence of metabolic endotoxemia [73] and bacteraemia [74], which, through an inflammatory response, influences insulin sensibility and, consequently, DM and metabolic complications.

Given the above, it is clear that a suitable GM and intestinal function is key for health maintenance and diabetes prevention, and it seems promising to tackle the diabetic disease through changes in GM composition and activity. In this light, a great number of studies have highlighted the therapeutic potential of microbiota-modulating dietary interventions in different forms of DM [75,76]. Although probiotics [77,78], prebiotics [79], and fermented foods [80,81,82] have served as a reference for microbiome-based interventions, postbiotics are emerging potential agents for DM prevention or management. Good evidence for this can be found in experimental studies in different models of diabetes mellitus [83,84] and few human trials [85], as discussed below.

## 2. Objectives and Search Strategy

The number of scientific articles examining postbiotic products for diabetes treatment or prevention is growing, which illustrates a considerable scientific interest in the topic. Nevertheless, to date, there is no review summarising the knowledge about this research topic. For this reason, the main objective of the present review was to provide insights into the currently available information on postbiotics in the context of DM, comprising microbial structures, metabolites, and inanimate microorganisms.

To perform a comprehensive review, we adopted the formalities of a systematic literature review. We conducted a search literature in MEDLINE (through PubMed) and Cochrane Library (CENTRAL) using key terms for postbiotic compounds (“bacteriocin”, “biosurfactant”, “cell-free extract”, “conjugated fatty acid”, “dead probiotic”, “enzyme”, “exopolysaccharides”, “extract”, “extracellular vesicles”, “flavonoid”, “GABA”, “gamma-aminobutyric acid”, “ghost probiotic”, “inactivate probiotic”, “lipoteichoic acid”, “muropeptides”, “neurotransmitter”, “oligosaccharides”, “paraprobiotic”, “peptide”, “peptidoglycan”, “phenolic”, “postbiotic”, “protein”, “SCFA”, “short-chain fatty acid”, “supernatant”, “s-layer protein”, “teichoic acid”, “terpenoid”, and “vitamin”) combined with terms for diabetes mellitus (“diabetes mellitus”, “type 1 diabetes”, and “type 2 diabetes”). Reference lists of included articles were also hand searched. We excluded faecal material transplants and other compounds or strategies that do not fit exactly with the postbiotic definition.

## 3. Results

Since a specific molecule may exert different physiological effects, we considered it appropriate to present and discuss findings according to their organic structures. In the following sections, we provide information regarding cell components, secreted compounds, and inanimate probiotics. In accordance with the new postbiotic definition [20], neither purified substances nor filtrates where cell components were removed could be classified as postbiotics. Nevertheless, the current literature available lacks enough studies for postbiotics in diabetes prevention or management, and we decided to include a few studies revolving around GABA and EVs.

### 3.1. Exopolysaccharides (EPSs)

EPSs are usually divided into homopolysaccharides (dextran, levan) or heteropolysaccharides (kefiran) made up of one or more types of monosaccharides, respectively [86]. Despite their being originally membrane components and participating in cell adhesion and protection, they differ from other membrane structures since they can be released to the environment and exert different functions [36] (reviewed in [52]). Microbial EPS has been extensively studied, especially in the food industry, where they significantly impact food properties (texture or stability), provide new sensory attributes, and improve the nutritional value of food [36]. Outstanding among the many beneficial properties associated with EPS is their prebiotic effect [36,52].

Our search yielded a total of six preclinical cross-sectional studies involving microbial EPS and diabetes models, and no human trials were found (Table 1). All the animal studies were performed on T1D models (alloxan or streptozotocin induced rodents), and a further work used an insulin-resistant HepG2 cell model, which is a human hepatocellular carcinoma cell line often used in diabetes research [87]. With the exception of one study [88], a decreased glucose level in the blood (blood fasting glucose, FBG) or supernatant was reported, and two studies [26,89] also observed an increase in insulin levels and an improved lipid profile. One study [83] reported that EPS from *B. licheniformis* was effective in counteracting oxidative stress and prevented diabetic complications by protecting key tissues and organs. One EPS from *L. plantarum* S1S2L2 resulted in the inhibition of pancreatic α-amylase in vitro [90], which is a treatment option contemplated for DM management. Other observed findings in this study include the upregulation of genes involved in glucose metabolism, the reduction of some metabolic end-products, and the increase of hepatic glycogen reservoirs.

While some EPSs caused a hypoglycaemic effect, another study failed to demonstrate any hypoglycaemic effect [88]. This illustrates the specific effect of postbiotics and evidences the need to indicate the characteristics of the studied molecule and the identity of the progenitor microorganism. Still, some studies did not specify the probiotic strain used [83,88,89], and this slows the progress of science. In addition, none of the studies mentioned above analysed the possible changes of the microbiota. 

Although we searched for other cell membrane elements, we did not identify any study involving oligosaccharides, s-layer proteins, teichoic acids, peptidoglycan, or muropeptides on DM.

### 3.2. GABA

GABA is a neurotransmitter and neuromodulator produced by β cells [92] that can also be produced by certain bacteria [93,94]. In addition to its important inhibitory effect in the central nervous system, increasing evidence has indicated that GABA can induce insulin production (insulinotropic effect), enhance glucose tolerance and insulin sensitivity, control the β cells mass, and exert anti-inflammatory and immunomodulatory effects, all of which are interesting in DM [92,95,96,97]. Indeed, an elegant study demonstrated that GABA signalling is compromised in T2D [98]. All this explains the growing interest in GABA in DM.

We identified two experimental studies examining the ability of GABA synthesised by probiotics to improve glucose metabolism and control glycaemia in rodent models of T1D and T2D (Table 2). The supplementation with yogurt fermented with a GABA-producing probiotic (*S. thermophiles* fmb5) in high-fat diet (HFD) and STZ-induced T2DM mice led to improved glucose tolerance and insulin sensitivity, probably due to a preserved pancreatic function and normalised organ function [95]. Besides the experimental product included remnants of probiotic cells, the beneficial properties were attributed to the GABA content.This intervention was ineffective in reducing FBG that, by contrast, was observed with a mung bean extract fermented with *Mardi rhizopus 5351* [84]. This mould displayed important antihyperglycaemic and antioxidant properties and also improved some components of the lipid profile in glucose- and alloxan-induced hyperglycaemic mice. Although GABA represents a promising approach, the existing literature does not indicate a strong antidiabetic or hypoglycaemic effect for this neuromodulator. We observed an important dose–response effect, and a greater effect was found with the highest GABA doses [84,94]. This feature must be considered in future studies.

### 3.3. Extracellular Vesicles (EVs)

EVs are spherical particles secreted by bacteria and other microorganisms, which discharge their cellular content, including proteins, polysaccharides, enzymes, and toxins among others. They enable the dialogue between microbial cells. Additionally, they can communicate with the host through microbe- and pathogen-associated molecular patterns (MAMP and PAMP, respectively), leading to an immunomodulatory action. The GM is a good source of EVs that move beyond the intestine and disseminate to other organs and tissues [44,99]. Furthermore, host cells also secrete EVs and recent evidence suggests that they could also play a role in the development and progression of T1D and T2D [100,101]. We identified a single study on EVs in which EVs from *A. municiphila* showed to improve the intestinal barrier function and glucose tolerance in HFD-induced T2D mice [40] (Table 2), and such effect was probably because of the observed changes in tight junction proteins, which prevented or reduced the risk of metabolic endotoxaemia. This study also demonstrated that EVs enhanced the barrier function in vitro experiments with Caco2 cells.

### 3.4. Supernatants, Extracts, and Surfactants

Our search yielded several studies on microbial supernatants, extracts, and surfactants, mainly on *Lactobacillus* and *Bacillus* probiotic species (See Table 3). Half of them were in vitro experiments and the other half animal studies in T1D- and T2D-induced mice. Controls varied from tap water [102] to phosphate-buffered saline [103] or commercial antidiabetic drugs [104,105].

Alpha-glucosidase inhibitors (AGI) are a group of antidiabetic drugs whose main function is reducing carbohydrate catabolism and thus minimising glucose availability and controlling glycaemic levels [106]. One study analysed the α-glucosidase inhibitory activity of five different *L. plantarum* strains’ cell-free supernatants, all of which demonstrated the ability to reduce glucose degradation, especially *L. plantarum* CCFM0236. The authors could confirm the beneficial effects of live *L. plantarum* CCFM0236 in a T2D murine model, resulting in a hypoglycaemic effect, improved insulin resistance and antioxidant capacity, preserved pancreatic function, and reduced inflammation [107]. Another study included eight LABs isolated from commercial water kefir grains, and the authors also evaluated cell-free extracts [108]. A similar study focused on the dipeptidyl peptidase IV (DPP-IV) inhibitory activity and the antioxidant activity of fourteen *Lactobacillus* strains isolated from traditional fermented foods [109]. DPP-IV is another pharmacological target for diabetes treatment [110]. The in vivo beneficial properties widely varied among the Lactobacillus strains, reinforcing the need to independently analyse each of the strains. The last work was conducted to investigate the antidiabetic effects of a soybean extract previously fermented with *B. subtilis* MORI [102]. The supplementation with a postbiotic derived from this microorganism prevented hyperglycaemia and oxidative stress in T1D animals but had no significant effects in healthy non-diabetic animals.

Biosurfactants are surface-active compounds with low or high molecular weight that help in a number of vital functions for cell homeostasis. Examples include lipoproteins and lipopeptides, glycolipids, fatty acids, lipopolysaccharides and heteropolysaccharides, and polymeric biosurfactants. Due to their properties, they present many technological applications (food industry, cosmetics, home care products, etc.) and could offer health benefits [111,112]. We found two studies evaluating the antidiabetic effect of *B. subtilis* SPB1 biosurfactant in T1D [104] and T2D [105]. The biosurfactant displayed hypoglycaemic effect and normalised serum α-amylase activity in both models, improved glucose tolerance in T2D animals, and improved the lipid profile and organ functions in T1D animals. These findings suggest that *B. subtilis* SPB1 biosurfactant may be a promising therapeutic agent for ameliorating different forms of DM. Finally, we identified one study that evaluated the in vivo effect of *L. rhamnosus* GG surfactant in a model of metabolic disorders [103]. The findings revealed that the treatment improved insulin sensitivity, protected from hyperlipidaemia, and also prevented hepatic steatosis in the animals.

**Table 3 foods-10-01590-t003:** Experimental evidence on the effects of microbial supernatants, extracts, and biosurfactants in diabetes mellitus.

Type of Research	Component	Bioactive Molecule(s)	Source/Origin	Model System (If Apply)	Findings	Study
Animal	Supernatant	Supernatant	*Lactobacillus rhamnosus* GG	C57BL/6J induced metabolic dysfunction with HFFD and intermittent hypoxia	↓ FBG (vs. baseline values), ↑insulin sensitivity, ↑ energy expenditure, improved body composition (fat and muscle mass), ↓ TC and TG, ↑ NEFAs, ↑ total faecal SFCAs, ↓ proinflammatory cytokines expression, downregulated lipogenesis, upregulated lipid oxidation	Liu et al., 2020 [103]
	Extract	Fermented soybean extracts (served dried with corn starch), low and high doses	*Bacillus subtilis* MORI/Isolated from Chungkookjang	Wistar rats induced T1D with STZ (55 mg/kg)	Low and high doses (vs. diabetic controls): ↑ BW, attenuated rise in FBG, ↓ food and water intake, ↓ MDA serum levels, ↑ CAT and GSH-Px activity, improved vascular function High doses (vs. diabetic control): ↑serum insulin levels and SOD levels	Lim et al., 2012 [102]
	Biosurfactant	Biosurfactant (served as crude lipopeptide preparation)	*Bacillus subtilis* SPB1/Isolated from Tunisian soil	Wistar rats induced T1D with alloxan (150 mg/kg)	↓ FBG and α-amylase activity in the plasma, ↓ TC, TG, and LDL levels, ↑ HDL levels, protected tissues (pancreatic b cells, liver, intestine, and kidney)	Zouari et al., 2015 [104]
				Wistar rats induced T2D with HFFD	↓FBG, improved glucose tolerance (OGTT), normalised serum α-amylase activity	Zouari et al., 2017 [105]
In vitro	Supernatant	CFS	*Five Lactobacillus plantarum* strains: CCFM0236, CCFM 12, CCFM 10, CCFM0311, and CCFM 23	-	α-glucosidase inhibitory activity (%): from 14.5 to 32.2	Li et al., 2016 [107]
		CFE and CFS	8 LAB isolates (K1, K8, K16, K19, K29, K35, K45, K96, and LGG)/Isolated from commercial water kefir grains	-	α-glucosidase inhibitory activity (%): from 5.2 to 39.4 in CFS, from 2.3 to 15.5 in CFE	Koh et al., 2018 [108]
		CFE, CFS and CFES	14 *Lactobacillus spp.* strains/Isolated from traditional fermented products	-	DPP-IV inhibitory activity (%): from 0 to 55.4 in CFE, from 0 to 7.13 in CFES/reducing activity (mmol of cysteine): from 73.3 to 189.7 in CFS, from 53.0 to 159.7 in CFE/DPPH free radical-scavenging activity (%): from 36.8 to 62.1 in CFS, from 12.9 to 34.5 in CFE/hydroxyl radical scavenging activity: from 13.7 to 68.6 in CFS; from 15.9 to 38.8 in CFE/superoxide anion radical scavenging activity (%): from 2.6 to 16.2 in CFS; from 12.2 to 43.3 in CFE/lipid peroxidation inhibiting capacity (%): from 1.5 to 18.5 in CFS; from 5.9 to 31.4 in CFE.	Yan et al., 2020 [109]

BW: body weight; CAT: catalase; CFS: cell-free supernatant; CFE: cell-free extract; CFES: cell-free excretory supernatants; DPP-IV: dipeptidyl peptidase IV; DPPH: 1,1-Diphenyl-2-Picryl-Hydrazyl; FBG: fasting blood glucose; GSH-Px glutathione peroxidase; HFFD: high-fat fructose diet; MDA: malondialdehyde; NEFAs: non-esterified fatty acid; OGTT: oral glucose tolerance test; SCFA: short-chain fatty acids; SOD: superoxide dismutase; UCP-1: uncoupling protein 1. ↑ means increase, ↓ means decrease

In the above-mentioned studies, the in vitro assays were performed with the purpose of identifying the strains that would, theoretically, exert the greatest hypoglycaemic effect in vivo and thus be regarded as antidiabetic probiotic candidates. Despite most of these supernatants and preparations not being intended for use as postbiotics, they exerted interesting antidiabetic effects, and more studies are needed to validate these findings.

With our search, we were able to identify a large number of research works performed on microbial-derived compounds, such as cell-wall muramyl-peptide [113], conjugated linoleic acids [114,115], or SCFAs [116,117,118], that were not obtained from a specific bacteria culture but were purchased commercially as a laboratory reagent. Although many of these products probably have a microbial origin, we decided not to include these scientific works since they do not contemplate the source and, as previously suggested, the identity of the producer microorganism has a significant influence on postbiotic functional properties.

### 3.5. Inanimate Microorganisms

Inanimate microorganisms, formerly referred to as paraprobiotics or ghost probiotics, have been extensively studied. They can be achieved by very different methods that will strongly determine their functional properties [119]. Although they are known to exert important immunomodulatory activities, they demonstrated benefits in very different conditions [37]. Our search yielded four experimental cross-sectional studies in murine models and two human trials (Table 4). The inactivation methods included heat, pressure, irradiation, and ohmic treatments. Only two studies [25,85] confirmed the absence of viable microbial cells in the preparations with inanimate preparations by culture-based analysis (plate count), and only two studies described changes in the GM [25,120]. A first study evaluated a live and inanimate multi-species probiotic and observed that both formulas were effective in reducing FBG, improving glucose tolerance, protecting pancreatic cells, and altering the intestinal tract modifying enteroendocrine cells, intestinal microbiota, and SCFA levels [25]. The authors brought together all the findings and concluded that the live probiotic was certainly more competent than the inanimate version, and it was hypothesised to be due to important changes in the intestinal environment, involving the inflammatory tone and the microbiota composition. In a second study, non-viable *B. longum* BR-108 presented a hypoglycaemic effect, reduced body weight gain and adiposity, and improved lipid profile in genetically obese mice [121]. The authors suggested that the inanimate probiotic could have provoked a hepatoprotective activity that explains such findings. Another study was performed in healthy rats fed with pasta enriched with either live or inanimated *B. animalis subsp. lactis* Bb-12 [120]. Both versions showed hypoglycaemic and hypocholesterolaemic effects as compared with the control group. The inanimation method (irradiation) barely changed the probiotic activity, and the only differences between the active and inanimate probiotic were related to the intestinal microbiota composition. The authors emphasised that the food matrix influences the action of the inanimate microorganism in a very significant manner, and therefore this aspect must be considered when discussing and extrapolating findings. The last study used the pasteurisation method to inactivate *A. muciniphila* and demonstrated that the postbiotic retained the beneficial effects described for the probiotic. Moreover, the pasteurised *A. muciniphila* improved certain parameters such as IR index, goblet cell density, normalisation of adipocyte diameter, and leptin levels in C57BL/6J mice fed with HFD diet [122].

We identified two human trials. The first study was a randomised and controlled crossover study, and it was performed in a reduced sample of healthy individuals; the findings indicate that the intake of live or inanimate *Lacticaseibacillus casei* 01 in a whey-grape juice with white bread can attenuate the glucose response expected [85]. Both formulas had an impact similar to the control drink (water), especially the inanimate probiotic preparation. These findings sustain the hypoglycaemic effect that was previously confirmed in a preliminary in vitro experiment. The second trial was a randomised double-blind placebo-controlled proof-of-concept and feasibility study using alive or pasteurised *A. municiphila.* Individuals with excess body weight (overweight or obese), insulin resistance, and a metabolic syndrome were enrolled and received placebo, live, or pasteurised *A. municiphila* (10^10^ bacteria per day) for 3 months. No adverse effects were observed, and the main results obtained were reduced plasma insulin levels, no differences in fasting blood glucose, and improvement of the insulin sensitivity index compared with the placebo group [123].

## 4. Discussion

The scientific literature examining the effects of postbiotics in DM is scarce compared with the number of publications focusing on probiotics, prebiotics, or fermented foods. Additionally, the majority of the included studies were published in the last ten years, and this may indicate that this research field is still in its infancy and that much knowledge remains to be discovered. Most eligible studies investigated compounds from probiotic bacteria, including Gram-positive (*Lactobacillus, Bacillus, Bifidobacterium,* and *Streptococcus*) and Gram-negative species (*Enterobacter, Sorangium*, and *Akkermansia*), and only one study evaluated a mould (*Rhizopus*) [84]. None of the studies considered postbiotics obtained from probiotic yeasts, even though they elicit many biological effects as well [24]. Although there were studies in different models of DM, studies on T1D outnumber studies on T2D.

The type of outcome measured widely differed among the different types of postbiotics. This is an expected result since several effects can be observed depending on the kind of molecule analysed. For example, according to previous study, it is likely that extracellular components exert metabolic activities, while cell components perform protective and antimicrobial functions [53]. Many studies have focused on key enzymes implicated in carbohydrate breakdown and metabolism. Previous studies on probiotics have demonstrated that some strains present an inhibitory effect on α-glucosidase activity [77,124], and the same was confirmed in EPS and cell-free supernatant. In the same line, supernatants from LABs also exhibited DPP-IV inhibitory activity, which was also described in fermented products [125]. At this point, scientists must consider postbiotics as a potential line of research for the development of natural AGI and DPP-IV inhibitors. In this sense, in a previous study one probiotic strain was genetically manipulated to produce β-lactoglobulin that included peptides with DPP-IV-inhibiting activity [126]. In the same line, another study modified one *E. coli* strain to synthesise N-acylphosphatidylethanolamines that showed protection against obesity-associated complications such as insulin resistance, adiposity or hepatosteatosis in animals [127]. This opens new opportunities for genetically modified organisms able to produce large amounts of compounds with biological effects that may be beneficial for DM management.

As introduced above, DM is frequently associated with leaky gut and gastrointestinal complications. Thus, other potential areas for future research may focus on intestinal permeability and integrity, as previously described in EVs from *A. municiphila* [40], intestinal peptides produced by modified probiotic bacteria [128], or soluble proteins such as p40 or p75 [20]. For example, one study identified a functional peptide from *S. epidermidis* JA1 supernatant that enhanced glucagon-like peptide-1 in NGN3-Human intestinal enteroid cells [129]. When the active bacteria was tested in vivo in HFD mice, it showed protection against metabolic disease. SCFAs, such as butyrate, propionate, and acetate, also have important stimulating effects in the intestine, where they nurture epithelial cells, contribute to mucin production, and participate in permeability regulation [3]. Indeed, they also induce the release of hormones and gastrointestinal peptides such as GLP-1, which aid in controlling glycaemia and glucose tolerance, mostly due to its insulinotropic effect [130,131]. One experiment showed the production of GLP-1 by a genetically modified *Lactococcus lactis subsp. Lactis* strain, and these novel bacteria demonstrated an insulinotropic effect in in vitro and in vivo experiments [128]. Similarly, a protein secreted from one *A. muciniphila* strain has been shown to induce GLP-1 secretion that resulted in improved metabolic condition in HFD mice [132]. Along the same line, Amuc_1100, a protein from the outer membrane of *A*. *muciniphila* MucT, improved the intestinal barrier function in the same model and is thought to mediate beneficial effects of the probiotic strain [122].

Inflammation is a common underlying factor in different forms of DM [64], and controlling intestinal and systemic inflammation is another alternative way to control DM. Since microbes and enteral immune cells coexist side by side, there is a very intimate connection between the GM and the immune system. Although mucosa tolerance is highly regulated, intestinal disturbances can drive a proinflammatory response, and today it is widely accepted that such activation can be instigated by PAMPs, such as LPS, above mentioned, and other cell components, such as peptidoglycans, flagellin, surface layer proteins, lipoproteins, or lipopolysaccharides, that are recognised by specific pathogen recognition receptors [77,133,134]. Accordingly, some postbiotic compounds likely exert immunomodulatory effects [37]; however, this has been less investigated in DM studies. Microbial-derived particles such as SCFAs could aid in controlling inflammation in DM by acting as immune regulators. Previous probiotic studies have shown that probiotics can drive benefits in diabetes and metabolic alternations by changing SCFA production [107,135,136]. The included studies, however, did not examine inflammatory markers, and only two studies, one on supernatants [103] and another on inanimate probiotics [25], evaluated cytokine levels and reported improvements in the inflammatory tone. Consequently, further studies should evaluate the effects of SCFAs and other postbiotic molecules on the immune and inflammatory response and diabetes progression. Lastly, postbiotics such as SCFAs [137] or EPSs [23] can modify the local microenvironment and lead to compositional and functional changes in the GM, which could improve glucose metabolism and control inflammation, thus causing benefits in DM.

In the future, researchers must provide a mechanistic basis for the effects of a given postbiotic, and the clinical benefits will ultimately be shown by well-designed double-blind, randomised, placebo-controlled trials. Human data on postbiotics and DM are lacking, and we identified only one trial on this topic in healthy adults [85]. It is likely that ethnic group [138,139], sex [140], and the time course of DM [141] influence the GM response to postbiotic treatments, and these aspects should be contemplated in human studies. Many of the included studies did not analyse the changes in the GM following the interventions, and this would provide valuable information about the underlying mechanisms. In addition, the long-term effects of postbiotics and whether a chronic exposition must be ensured to preserve their beneficial effects remains to be studied. Probably, this will vary among the different postbiotic compounds.

The lack of adverse events is a clear advantage over classic pharmacological antidiabetic drugs that normally have side effects [142]. Nevertheless, we do not know yet whether postbiotics can provide the same benefits as current antidiabetic strategies or if they could be used in combination with classic antidiabetic drugs. As mentioned above, the identity (up to strain level) of the progenitor microorganisms, as well as the production conditions used for the postbiotic manufacturing, strongly impact postbiotics’ properties. This limits the ability to compare different experiments and to generalise the findings. In our review, we found many articles that ignored the strain of the microorganisms used in the experiments, and we invite other research teams working on postbiotics to carry out a detailed characterisation of the microbial strains they are working on. In the same line, the different delivery matrices (yogurt [95], pasta [120], liquid solution [89]), may have a key impact on postbiotics’ biological effects.

To date, very little research has focused on the antidiabetic effects of SCFAs, phenolic compounds, vitamins, peptides, or bacteriocins, and therefore there is no information about their effects on DM, and future studies are warranted. Along the same line, we did not detect any investigation on vitamins or phenolic-derived postbiotics. A previous in vitro study indicated that tannic acid exerts α-glucosidase and α-amylase inhibitory effects and could therefore offer health benefits in DM [143]. Regarding the bacteriocins, we found one bio-gel prepared with the bacteriocin nisin as an antimicrobial compound, which was effective in alleviating diabetic food infections along with conventional antibiotic treatment [144]. Finally, although less conventional, faecal material transplants have provoked intense interest [145], and human studies have demonstrated improvements in insulin resistance and in faecal metabolites that were attributed to changes in GM composition and activity [146,147]. Thus, they also offer promising opportunities for DM management.

## 5. Conclusions

To the best of our knowledge, this is the first review attempting to summarise the evidence for postbiotics’ effects on DM. Although expertise in this area is still rather limited and there are inconsistencies between the results of the studies, the available literature suggests that this new entity in the -biotics field opens the door to new therapeutic and preventive approaches for DM and other metabolic diseases. The analysed literature focuses on how aspects such as the progenitor microorganism, the matrix, and the fermentation conditions, affect the biological effects of postbiotics in a relevant manner. The lack of adverse events and the interesting outcomes reported in the included studies should encourage further studies on postbiotics for the prevention and alleviation of DM. This new field of research represents a great opportunity for researchers, doctors, biotechnologists, and food technicians to join forces in the use of postbiotics as novel therapeutic candidates for the treatment of diseases such as DM.

## Figures and Tables

**Figure 1 foods-10-01590-f001:**
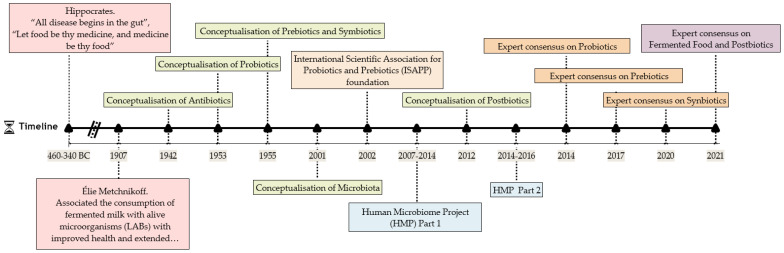
Timeline illustrating the main historical milestones in microbiology, the biotics family and health.

**Figure 2 foods-10-01590-f002:**
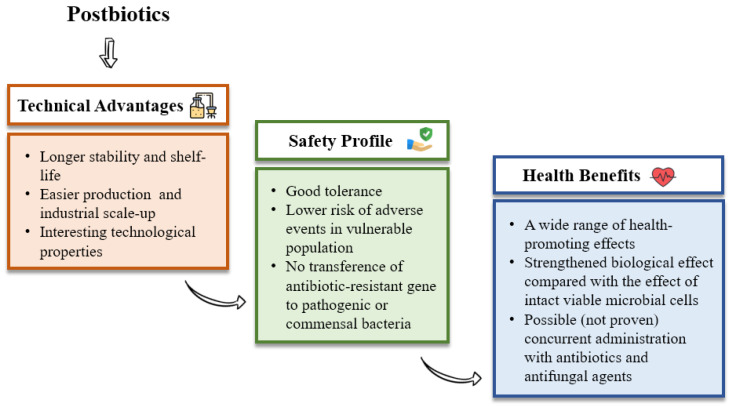
General characteristics of postbiotics.

**Figure 3 foods-10-01590-f003:**
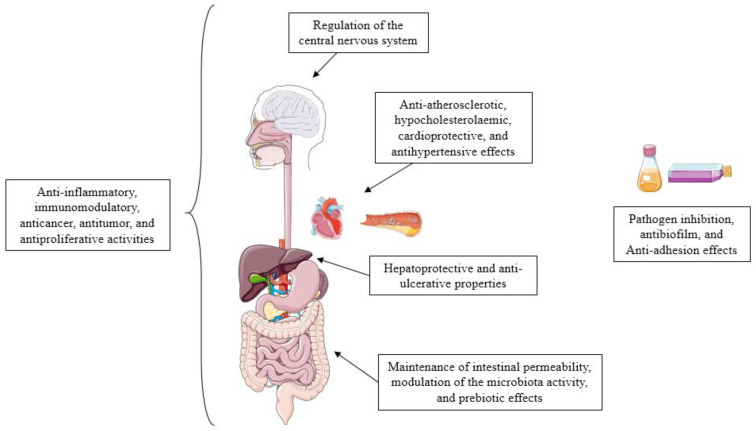
Potential mechanisms of action of postbiotics (illustrations from Smart^®^).

**Table 1 foods-10-01590-t001:** Experimental studies on microbial exopolysaccharides in diabetes mellitus.

Type of Research	Bioactive Component	Source/Origin	Model System	Main Findings	Study
Animal	Levan	*Bacillus licheniformis*	Wistar rats induced T1D with alloxan (150 mg/kg BW)	↓Glycaemia, ↑glycogen level, ↓AST, ALT, bilirubin, creatinine, and urea levels	Dahech et al., 2011 [83]
	Levan	*Bacillus subtilis* (Natto)	Wistar rats induced T1D with STZ (65 mg/kg BW)	No hypoglycaemic effect. No improvement of diabetes symptoms	Bazani et al., 2012 [88]
	Exopolysaccharide (unspecified)	*Bacillus subtilis*	Sprague-Dawley rats induced T1D with STZ (65 mg/kg BW)	↓FBG, ↑serum insulin levels, ↓TC, LDL, VLDL and TG, ↑HDL in treated vs. control rats	Ghoneim et al., 2016 [89]
	Selenium-enriched exopolysaccharide	*Enterobacter cloacaceae* Z0206	Female ICR mice induced T1D with alloxan (190 mg/kg)	↓FBG, ↑serum insulin level, ↓glycosylated serum protein, ↑BW, ↓TC and TG in treated vs. control mice	Jin et al., 2012 [26]
	Exopolysaccharide (unspecified)	*Sorangium cellulosum NUST06*	Mice (Kunming strain) induced T1D with alloxan (250 mL/kg BW)	↓FBG in both healthy and alloxan-induced diabetic mice	Ding et al., 2004 [91]
Cell line	Exopolysaccharide (unspecified)	*Lactobacillus plantarum* H31-2	In vitro, insulin-resistant HepG2 cells	↓Supernatant glucose concentration of insulin-resistant HepG2 cells, inhibition of pancreas α-amylase, upregulation of the expression of GLUT-4, Akt-2, and AMPK	Huang et al., 2020 [90]

ALT: alanine aminotransferase; AST: aspartate aminotransferase; BW: body weight; FBG: fasting blood glucose; ICR: Institute Cancer Research; HDL: high-density lipoprotein; LDL: low-density lipoprotein; STZ: streptozotocin; TC: total cholesterol; TG: triglyceride; T1D: type 1 diabetes; VLDL: very-low-density lipoprotein. ↑ means increase, ↓ means decrease.

**Table 2 foods-10-01590-t002:** Experimental studies on the effects of GABA and EVs in diabetes mellitus.

Compound	Source/Origin	Model System	Main Findings	Study
GABA	GABA-containing fermented mung bean extract with *Mardi rhizopus* 5351 inoculums, low and high doses	T1D Balb/c induced T1D with high doses of alloxan (100 mg/kg BW) and glucose-induced hyperglycaemic Balb/c mice	Glucose-induced mice: ↓FBG Alloxan-induced mice: ↓ Reduced FBG, ↑insulin serum levels, ↓ TC and TG serum levels (vs. nonfermented), restored antioxidant status (↓MDA and NO)	Yeap et al., 2012 [84]
	Yogurt fermented with *Streptococcus thermophiles* fmb5, low and high doses	C57BL/6 mice induced T2D with low doses of STZ (100 mg/kg BW)	Low and high doses: No hypoglycaemic effect, ↑HOMA-β, improved glucose tolerance and insulin resistance, normalised fat, kidney and liver coefficients, ↓serum urea nitrogen, no effect on HbA1c nor BW. High doses: Normalised pancreatic histology, preserved islet cells function, ↓ TC and LDL, HDL	Li et al., 2020 [95]
EVs	*Akkermansia muciniphila* ATCC BAA-835	C57BL/6 mice induced T2D with HFD	In vivo: Reduced BW, attenuated intestinal damage following HFD, increase expression of tight junction proteins, improved glucose tolerance (OGTT) In vitro: Enhanced barrier function in Caco-2 cells	Chelakkot et al., 2018 [40]

AUC: area under the curve; BW: body weight; EVs: extracellular vesicles; FBG: fasting blood glucose; GABA: γ-aminobutyric acid; GIP: gastric inhibitory polypeptide; GLP-1: glucagon-like peptide-1; GM: gut microbiota; HFD: high fat diet; HOMA: homeostasis model assessment-β; LDL: low-density lipoprotein; MDA: malondiadehyde; NO: nitric oxide; OGTT: oral glucose tolerance test; PYY: peptide YY; STZ: streptozotocin; TC: total cholesterol; TG: triglyceride; T1D: type 1 diabetes; T2D: type 2 diabetes. ↑ means increase, ↓ means decrease.

**Table 4 foods-10-01590-t004:** Experimental and clinical studies on the effects of inanimate probiotics in diabetes mellitus and health.

Type of Research	Microorganism	Inactivation Method	Model System	Main Findings	Study
Animal	*Lactobacillus casei* CCFM419, *L*. *plantarum* X1, *L*. *rhamnosus* Y37, *L. brevis* CCFM648, and *L. plantarum* CCFM36	Heat treatment (80 °C for 30 min)	C57BL/6J mice induced T2D with STZ (100 mg/kg BW)	Inanimate probiotic: ↑ Serum IL-6 levels and faecal acetic levels Live probiotic: Improved insulin tolerance, normalised serum IL-10, TNF-α and IL-6 levels, ↑faecal acetic and butyrate levels, ↑faecal *Lactobacillus*, *Akkermansia*, and *Bifidobacterium* genera, ↑faecal actinobacteria (%) Both: ↓ FBG, normalisation of HbA1c and leptin levels, ↑ileum L cell levels, ↑faecal Firmicutes/Bacteroidetes ratio Both with stronger effect with live probiotics: Improved glucose tolerance, protection of pancreatic histological characteristics	Li et al., 2016 [25]
	*Bifidobacterium longum* BR-108; low, medium, and high doses	Heat and pressure treatment (autoclaved at 105 °C for 20 min)	Tsumura Suzuki obese diabetes (TSOD) mice (genetically obese mice)	All the doses: ↓BG gain (vs. control mice), ↓adipose tissue accumulation, no differences in food consumption, ↓serum creatinine levels Medium and high doses: ↓FBG, ↓NEFAs, ↓creatinine urine levels Medium dose: ↓TC High doses: Improved glucose tolerance (OGTT), ↓TG	Ben Othman et al., 2019 [121]
	*Bifidobacterium animalis* subsp. lactis Bb-12, incorporated in wheat pasta	Irradiation (gamma-irradiation on ice, at 2.5 Kilogray)	Healthy Wistar rats	Control and paraprobiotic pasta: ↓FBG and TC (vs. control diet), no differences in food consumption and BW, TG, HDL, AST, ALT, and microbiota alpha-diversity indexes Paraprobiotic pasta: Differential microbiota composition (vs. control diet and control pasta)	Almada et al., 2021 [120]
	*Akkermansia muciniphila*	Pasteurised (70 °C for 30 min)	C57BL/6J mice (normal chow or high-fat diet)	↓ IR index, ↑ faecal caloric content, ↑ goblet cell density, normalisation of adipocyte diameter and ↓leptin levels (vs. live microorganism)	Plovier et al., 2017 [122]
Human	*Lacticaseibacillus casei* 01; in whey-grape juice drink	Ohmic heating (8 V/cm, 95 °C/7 min, 60 Hz)	In vitro experiments; healthy volunteers (n = 15)	Preliminary in vitro experiments: Live and inanimate probiotic had α-glucosidase and α-amylase inhibitory activities Postprandial glucose levels in healthy volunteers: Accelerated increase in PBG with the probiotic and inanimate probiotic drinks (vs. control) due to differences in sugar content with the control (water), no differences in AUC values, inanimate probiotic had a similar effect to control.	Barros et al., 2021 [85]
	*Akkermansia muciniphila*	Pasteurised	Volunteers with excess body weight (overweight or obese), insulin resistance and a metabolic syndrome	↓ insulin levels, improved insulin sensitivity index, fasting glycaemia and HbA1c were not modified (vs. placebo group)	Depommier et al., 2019 [123]

ALT: alanine aminotransferase; AST: aspartate aminotransferase; AUC: area under the curve; BW: body weight; FBG: fasting blood glucose; HbA1c: glycated haemoglobin; HDL: high-density lipoprotein; IR: insulin resistance index; NEFAs: non-esterified fatty acid; OGTT: oral glucose tolerance test; PBG: postprandial blood glucose; STZ: streptozotocin; TC: total cholesterol; TG: triglyceride; T1D: type 1 diabetes; T2D: type 2 diabetes.

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
