# Peer review of "Role of Postbiotics in Diabetes Mellitus: Current Knowledge and Future Perspectives"

_foods, 2021, doi:10.3390/foods10071590_

Round 1

Reviewer 1 Report

The work by Cabello-Olmo et al. extensively reviewed evidences about the use of postbiotics in diabetes. Although the topic is interesting, there are some major issues to be addressed:

-the introduction part should not start directly with postbiotics. In addition, even if the paragraph is named "postbiotics", the authors introduce in a confusing way the concepts of probiotics, prebiotics, symbiotic, as well as the role of the gut microbiome. I suggest to split the paragraph in:

  • brief introduction on the gut microbiome, dysbiosis and the role on health/disease with focus on diabetes
  • separate paragraph on probiotics/prebiotics and evidence on their role on gut microbiota and disease

-the author suggest that fermented foods may be considered postbiotics (lines 147-158). this concept should be expanded more, e.g. evidence from studies on the role of FFs on health and gut microbiome (see https://academic.oup.com/femsre/article/44/4/454/5859486)

-from paragraph 3.1 on, the authors describe the effect of different molecules of microbial origin on health. it may be interesting adding more information of the tables: if the study monitored the gut microbiome changes, the effects on the gut microbiome, the use of a control group, the type of placebo used by the control group, the study design (e.g., cross-sectional or other)

-in table 2, a study using a GABA producing strain in yogurt fermentation was included. How the role of GABA was spliced from the role of the strain? it seems a probiotic study ore than a postbiotic... this concern should be at least highlighted in the text

-paragraph 3.5: some important studies on novel probiotics/postbiotics are missing and should be discussed, e.g. https://www.nature.com/articles/nm.4236, https://www.ncbi.nlm.nih.gov/pmc/articles/PMC6699990/

-the discussion is quite superficial and should address the topic in a more critical view. Some specific points can be discussed: role of strain level diversity in probiotics/postbiotics; impossibility to compare the results in literature since different studies used different microbial species/strains and different study design; role of personalized features of the gut microbiome in response to probiotics/postbiotics 

Author Response

We very much appreciate the prompt consideration of our manuscript and constructive criticism of the reviewer. Detailed below is a summary of the changes that have been made in the manuscript with appropriate rebuttal comments. Changes to the manuscript addressing point-by-point reviewer’ comments are itemized below. We hope that this improved manuscript satisfy the requests and is acceptable for publication in Foods.

Reviewer #1:

Comment 1: the introduction part should not start directly with postbiotics. In addition, even if the paragraph is named "postbiotics", the authors introduce in a confusing way the concepts of probiotics, prebiotics, symbiotic, as well as the role of the gut microbiome. I suggest to split the paragraph in:

  • brief introduction on the gut microbiome, dysbiosis and the role on health/disease with focus on diabetes
  • separate paragraph on probiotics/prebiotics and evidence on their role on gut microbiota and disease

We agree with the reviewer that the title “1.1. Postbiotics” confuses the distribution of the information in the Introduction section; so, we have deleted the title (line 30 previous version) and we have move it after the Figure 1. We hope that the change introduced will improve the understanding.

Comment 2: the author suggest that fermented foods may be considered postbiotics (lines 147-158). this concept should be expanded more, e.g. evidence from studies on the role of FFs on health and gut microbiome (see https://academic.oup.com/femsre/article/44/4/454/5859486)

We thank the reviewer for pointing this out. After carefully considering this suggestion and rereading our manuscript, we realised that we have not expressed ourselves clearly enough in our text. We did not intend to suggest that fermented foods are within the group of postbiotics and we apologize for this. Before writing the present review we have conscientious read and studied the International Scientific Association for Probiotics and Prebiotics (ISAPP) updated the definition for postbiotics (Salminen et al., 2021) and fermented foods (Marco et al.,2021), and we are aware of the minor but significant differences between the two categories. In agreement with the expert consensus, we believe that only certain fermented foods would meet the criteria for postbiotics (i.e: when the postbiotic derives from a fermented product made using a defined microorganism or microbial consortia). In this review we have focused on postbiotics, and besides we reckon that other related terms, such as probiotics, prebiotics, or fermented foods are important but we decided not to included excesive information on them since this may confuse readers. To avoid any misunderstandings and to address the reviewer’s concern, we have modified the manuscript (see comments on the right side of the manuscript). We hope the reviewer and the editor will follow us in our argumentation here.

Comment 3: from paragraph 3.1 on, the authors describe the effect of different molecules of microbial origin on health. it may be interesting adding more information of the tables: if the study monitored the gut microbiome changes, the effects on the gut microbiome, the use of a control group, the type of placebo used by the control group, the study design (e.g., cross-sectional or other)

We agree with the reviewer that more information regarding the studies’ design should be provided. We have included more information regarding the study design and the analyzed parameters in the included studies in section “3. Results”. The added information can be found in the main text together with the corresponding comments on the right side of the manuscript.

Comment 4: in table 2, a study using a GABA producing strain in yogurt fermentation was included. How the role of GABA was spliced from the role of the strain? it seems a probiotic study more than a postbiotic... this concern should be at least highlighted in the text.

It is true that the study of Li et al. 2020 relies on a product elaborated with viable microorganisms (Streptococcus thermophilus fmb5), however, the article does not provide information on the amount of probiotics used for the inoculation nor the number of colony forming units in the final product. In addition, the objective of the study was to assess the effect of GABA-yogurt in a murine diabetic model, and, as stated in the discussion and conclusion, the authors suggest that the observed effects were because of the GABA content. Taking this in consideration, along with the fact that the authors stored the experimental yogurt at -80ºC until use, we assume that the load of viable microorganisms in the GABA-yogurts is negligible and may not be responsible for the observed effects. Besides this, we agree with the reviewer that this aspect should be mentioned in the text, and we have therefore included a sentence in the main text together with the corresponding comments on the right side of the manuscript.

Comment 5: paragraph 3.5: some important studies on novel probiotics/postbiotics are missing and should be discussed, e.g. https://www.nature.com/articles/nm.4236, https://www.ncbi.nlm.nih.gov/pmc/articles/PMC6699990/

We appreciate the reviewer’s indication to include the novel studies conducted with A. muciniphila. We have included the studies in the 3.5 paragraph, in table 4 and also in different parts in the discussion section (see comments on the right side of the manuscript regarding the modifications done).

Comment 6: the discussion is quite superficial and should address the topic in a more critical view. Some specific points can be discussed: role of strain level diversity in probiotics/postbiotics; impossibility to compare the results in literature since different studies used different microbial species/strains and different study design; role of personalized features of the gut microbiome in response to probiotics/postbiotics.

We appreciate the reviewer’s suggestion for a more exhaustive discussion. The group of postbiotics is very heterogeneous and thus it was very complex to elaborate a discussion embracing all the studied compounds and debating the possible mechanism behind the observed effects. In this case, this would be too long. For that reason, we have elaborated a more general but still inclusive discussion. We made a few changes in the discussion section in order to accommodate the reasonable suggestion made by the reviewer. The added information can be found in the main text together with the corresponding comments on the right side of the manuscript.

Once again, we thank you for the time you put in reviewing our paper and look forward to meeting your expectations. We look forward to hearing from you in due time regarding our submission and to respond to any further questions and comments you may have.

Sincerely,

Miguel Barajas

Reviewer 2 Report

The reviewed paper concerns role of postbiotics in diabetes mellitus. Largely, the work performed is of good quality and conclusions are supported by the data.  It is really interesting and important to know potential therapeutic strategy for diabetes mellitus, but  need to be corrected. 

Below, there is a list of suggestions that in my opinion would help to improve the manuscript.

1.    At present your manuscript is not of sufficient English quality to be moved forward. 
2.    Please make the Figure 1 simpler
3.    Please pay attention to the literature 

Author Response

We very much appreciate the prompt consideration of our manuscript and constructive criticism of the reviewer. Detailed below is a summary of the changes that have been made in the manuscript with appropriate rebuttal comments. Changes to the manuscript addressing point-by-point reviewer’ comments are itemized below. We hope that this improved manuscript satisfy the requests and is acceptable for publication in Foods.

Reviewer 2:

Comment 1: At present your manuscript is not of sufficient English quality to be moved forward.

We appreciate the constructive comment to the reviewer. In this sense, we have made changes to the grammar of the text. In addition, we have asked the publisher for help with the final revision of the manuscript, thus ensuring that the final version will not have any typographical or grammar errors.

Comment 2: Please make the Figure 1 simpler.

We agree that Figure 1 was overloaded with too many details and we have modified it accordingly. We have removed all the illustrations, we have simplified the sentences on the timeline, and we have reduced the size of the figure. We believe that this new version of the figure is much clearer and, as the reviewer indicates, it now focuses on the message we want to send.

In any case, if the final version of the figure does not meet the reviewer's expectations, please do not hesitate to tell us in order to improve it.

Comment 3: Please pay attention to the literature 

We have followed the reviewer’s recommendation and we have extensively reviewed the literature in the whole manuscript. We have added updated references and modified those that, as suggested by the reviewer, were not according to the information we wanted to highlight. In this sense, the new version of the manuscript has improved considerably in relation to the cited literature.

Once again, we thank you for the time you put in reviewing our paper and look forward to meeting your expectations. We look forward to hearing from you in due time regarding our submission and to respond to any further questions and comments you may have.

Sincerely,

Miguel Barajas

Round 2

Reviewer 1 Report

the authors took into account all suggestions